# Phase 1/2 Trial of CLAG-M with Dose-Escalated Mitoxantrone in Combination with Fractionated-Dose Gemtuzumab Ozogamicin for Newly Diagnosed Acute Myeloid Leukemia and Other High-Grade Myeloid Neoplasms

**DOI:** 10.3390/cancers14122934

**Published:** 2022-06-14

**Authors:** Colin D. Godwin, Eduardo Rodríguez-Arbolí, Megan Othus, Anna B. Halpern, Jacob S. Appelbaum, Mary-Elizabeth M. Percival, Paul C. Hendrie, Vivian G. Oehler, Siobán B. Keel, Janis L. Abkowitz, Jason P. Cooper, Ryan D. Cassaday, Elihu H. Estey, Roland B. Walter

**Affiliations:** 1Clinical Research Division, Fred Hutchinson Cancer Center, Seattle, WA 98109, USA; cgodwin@fredhutch.org (C.D.G.); earboli@fredhutch.org (E.R.-A.); halpern2@uw.edu (A.B.H.); jappelba@fredhutch.org (J.S.A.); mperciva@uw.edu (M.-E.M.P.); phendrie@uw.edu (P.C.H.); voehler@uw.edu (V.G.O.); sioban@uw.edu (S.B.K.); janabk@uw.edu (J.L.A.); jasonc8@uw.edu (J.P.C.); cassaday@seattlecca.org (R.D.C.); 2Division of Hematology, Department of Medicine, University of Washington, Seattle, WA 98195, USA; 3Public Health Sciences Division, Fred Hutchinson Cancer Center, Seattle, WA 98109, USA; mothus@fredhutch.org; 4Department of Laboratory Medicine & Pathology, University of Washington, Seattle, WA 98195, USA; 5Department of Epidemiology, University of Washington, Seattle, WA 98195, USA

**Keywords:** acute myeloid leukemia (AML), adults, antibody-drug conjugate, chemotherapy, CD33, CLAG-M, gemtuzumab ozogamicin, phase 1/2 trial

## Abstract

**Simple Summary:**

Several studies have demonstrated that gemtuzumab ozogamicin (GO) improves outcomes with intensive chemotherapy in some adults with acute myeloid leukemia (AML), but it has remained unclear which dosing schedule of GO is best. Here, we conducted a phase 1/2 study in 66 adults with newly diagnosed AML or other high-grade myeloid neoplasm, and found that a fractionated dosing schedule of GO (GO3) can be safely combined with cladribine, high-dose cytarabine, G-CSF, and dose-escalated mitoxantrone (CLAG-M). Fifty-two out of sixty (87%) patients treated with CLAG-M/GO3 achieved a complete remission (CR)/CR with incomplete hematologic recovery (CRi), 45/52 (87%) without flow cytometric measurable residual disease. Six- and twelve-month event-free survival were 73% and 58%; among favorable-risk patients, these estimates were 100% and 95%. Compared to 186 medically matched adults treated with CLAG-M alone, CLAG-M/GO3 was associated with better survival in patients with favorable-risk disease. These data indicate that CLAG-M/GO3 is safe and more efficacious than CLAG-M alone in favorable-risk AML/high-grade myeloid neoplasm.

**Abstract:**

Gemtuzumab ozogamicin (GO) improves outcomes when added to intensive AML chemotherapy. A meta-analysis suggested the greatest benefit when combining fractionated doses of GO (GO3) with 7 + 3. To test whether GO3 can be safely used with high intensity chemotherapy, we conducted a phase 1/2 study of cladribine, high-dose cytarabine, G-CSF, and dose-escalated mitoxantrone (CLAG-M) in adults with newly diagnosed AML or other high-grade myeloid neoplasm (NCT03531918). Sixty-six patients with a median age of 65 (range: 19–80) years were enrolled. Cohorts of six and twelve patients were treated in phase 1 with one dose of GO or three doses of GO (GO3) at 3 mg/m^2^ per dose. Since a maximum-tolerated dose was not reached, the recommended phase 2 dose (RP2D) was declared to be GO3. At RP2D, 52/60 (87%) patients achieved a complete remission (CR)/CR with incomplete hematologic recovery (CRi), 45/52 (87%) without flow cytometric measurable residual disease (MRD). Eight-week mortality was 0%. Six- and twelve-month event-free survival (EFS) were 73% and 58%; among favorable-risk patients, these estimates were 100% and 95%. Compared to 186 medically matched adults treated with CLAG-M alone, CLAG-M/GO3 was associated with better survival in patients with favorable-risk disease (EFS: *p* = 0.007; OS: *p* = 0.030). These data indicate that CLAG-M/GO3 is safe and leads to superior outcomes than CLAG-M alone in favorable-risk AML/high-grade myeloid neoplasm.

## 1. Introduction

Several randomized controlled trials have tested the addition of the CD33 antibody-drug conjugate gemtuzumab ozogamicin (GO) to intensive chemotherapy for adults with newly diagnosed acute myeloid leukemia (AML) [1,2,3,4,5,6,7]. A meta-analysis of data from 3325 patients enrolled on five of these trials showed that GO reduced relapses and improved relapse-free survival (RFS) and overall survival (OS) [8]. Although now approved in many countries, how best to combine GO with intensive chemotherapy remains uncertain because the trials differed in patient characteristics, chemotherapy backbones, and GO doses/schedules. In the above meta-analysis, the largest point estimate of benefit from GO was seen in the ALFA-0701 trial, a trial using 7 + 3 together with fractionated doses of GO (3 mg/m^2^ [capped at 1 vial or 5 mg] on days 1, 4, and 7 [GO3]) rather than a single dose GO [8]. Whether GO3 can be safely combined with high-dose cytarabine-based chemotherapy instead of 7 + 3 is unknown. To address this, we conducted a single center, single arm phase 1/2 trial of GO with cladribine, high-dose cytarabine, G-CSF, and escalated-dose mitoxantrone (CLAG-M). Because we found CLAG-M to be safe in medically fit adults with newly diagnosed AML and to induce high rates of remissions, with a high proportion of these remissions being without evidence of measurable residual disease (MRD) [9,10], this backbone is currently routinely used as intensive AML induction chemotherapy at our institution.

## 2. Patients and Methods

### 2.1. Study Population

Adults ≥18 years with untreated AML [11] (except acute promyelocytic leukemia) or other high-grade myeloid neoplasm (as defined by the presence of ≥10% blasts in either the blood and/or the bone marrow) were eligible regardless of CD33 expression on AML blasts or CD33 single nucleotide polymorphism rs12459419 [7], provided they had a treatment-related mortality (TRM) score of ≤13.1. As described before, weighted information from eight covariates is used to compute this score (online calculator: https://trmcalculator.fredhutch.org (accessed on 13 June 2022)): age, performance status, white blood cell [WBC] count, peripheral blood blast percentage, type of AML [de novo vs. secondary], platelet count, albumin, and creatinine) [12]. A TRM score of ≤13.1 corresponds to a ≤13.1% probability of mortality within 28 days of initiating intensive chemotherapy for newly diagnosed AML [12]. In the cohort of 2238 patients in whom the TRM score was originally derived, 80% had a score of 0–13.1, whereas 20% had a score of 13.1–100 [12]. Patients were also required to have a left ventricular ejection fraction (LVEF) ≥45%, a serum creatinine ≤2.0 mg/dL, and a total bilirubin ≤2.5 times the upper limit of normal. Additionally, they could not have any uncontrolled infection and there needed to be an expected survival of >1 year absent AML. Prior low-intensity treatment (e.g., azacitidine/decitabine, lenalidomide, growth factors) for lower-grade myeloid neoplasm (<10% blasts) was permitted. The Fred Hutchinson Cancer Center (Fred Hutch) Institutional Review Board (IRB) approved the protocol (ClinicalTrials.gov: NCT03531918 [https://clinicaltrials.gov/ct2/show/NCT03531918 (accessed on 13 June 2022)]), and written informed consent was provided by patients in accordance with the Declaration of Helsinki.

### 2.2. Disease and Response Classification

The refined MRC/NCRI [13] and the 2017 European LeukemiaNet (ELN) criteria [14] were used to assess disease risk. Secondary disease was defined either as AML transformed from antecedent hematologic disorder, high-grade myeloid neoplasm other than AML if there was a history of lower-grade myeloid neoplasm (i.e., <10% in blood or marrow), or AML/MDS in a patient who had previously received cytotoxic therapy or radiation. Best responses were assessed after 1–2 cycles of induction chemotherapy and defined according to standard criteria [14]. Measurable residual disease (MRD) was assessed by multiparametric flow cytometry (MFC) as described, with any level of residual disease considered MRD^pos^ [15,16,17,18]. Relapse after study treatment was defined by standard morphologic criteria [14] or any disease recurrence at the MRD or cytogenetic level if leading to therapeutic intervention.

### 2.3. Treatment Plan

Based on the data from a previous institutional phase 1/2 trial [9,10], CLAG-M with escalated-dose mitoxantrone was used as chemotherapy backbone. This regimen consists of intravenous (IV) cladribine at 5 mg/m^2^/day (days 1–5), IV cytarabine at 2 g/m^2^/day (days 1–5), subcutaneous G-CSF at 300 or 480 μg/day (for weight <76 kg vs. ≥76 kg; days 0–5), and IV mitoxantrone 18 mg/m^2^/day (days 1–3). If the WBC count was >20,000/μL, the first two doses of G-CSF could be omitted. Patients were assigned CLAG-M with either one dose of GO (“GO1”: 3 mg/m^2^ on day 1) or GO3 (3 mg/m^2^ on days 1, 4 and 7) in phase 1. All doses of GO were capped at the current vial content of 4.5 mg per the U.S. prescribing package insert (https://www.accessdata.fda.gov/drugsatfda_docs/label/2017/761060lbl.pdf (accessed on 13 June 2022)). In phase 2, patients received CLAG-M/GO at the recommended phase 2 dose (RP2D) identified in phase 1. A second course of CLAG-M without GO was given if CR without MRD, a response with particularly favorable outcome in a retrospective institutional analysis [19], was not achieved with cycle 1. Post-remission therapy for patients achieving a CR or CR with incomplete hematologic recovery (CRi) with 1–2 courses of induction therapy included CLAG (i.e., mitoxantrone omitted) for one cycle and high-dose cytarabine for two additional cycles. No GO was given during post-remission therapy. Continuation on protocol-defined post-remission therapy vs. study discontinuation and consolidation with hematopoietic cell transplantation (HCT) or other regimen was at the discretion of the treating physician. A protocol amendment was implemented in August 2019 after the first 32 patients were enrolled to allow FLT3 inhibitor therapy for patients with FLT3-mutant AML; this was used in three patients. Patients were removed from study therapy for failure to achieve CR/CRi after two treatment cycles, use of alternative post-remission therapy, excess toxicity (including persistent aplasia without evidence of leukemia after day 42 of treatment), or relapse. Toxicities were evaluated based on the CTCAE (NCI Common Terminology Criteria for Adverse Events) Version 5.0 (http://ctep.cancer.gov (accessed on 13 June 2022)).

### 2.4. Comparison of CLAG-M + GO with CLAG-M

For non-randomized comparison, we identified adults ≥18 years treated with CLAG-M with escalated-dose mitoxantrone at 18 mg/m^2^/dose for newly diagnosed non-APL AML or other high-grade myeloid neoplasm at our institution, some of whom had been treated as part of a previously reported phase 1/2 study [9]. For medical fitness matching, patients were required to have a TRM score of ≤13.1, serum creatinine ≤2.0 mg/dL, total bilirubin ≤2.5 times the upper limit of normal, and LVEF ≥45% if cardiac function measurements were available.

### 2.5. Statistical Considerations

In phase 1 of the CLAG-M/GO trial, cohorts of six patients were enrolled (“rolling-six design”). Dose-limiting toxicities (DLTs) were assessed during cycle #1 and were defined as: (1) any grade 3 non-hematologic toxicity, other than febrile neutropenia or infection, lasting >48 h that resulted in a >7-day delay of subsequent treatment; (2) any grade ≥4 non-hematologic toxicity, other than constitutional symptoms if recovery to grade ≤2 within 14 days or febrile neutropenia or infection. The highest dose studied in which the incidence of DLTs was ≤33% was defined as the maximum tolerated dose (MTD). Escalation to GO3 was possible if ≤1/6 patients treated with GO1 experienced a DLT. If ≤2/6 patients treated with GO3 experienced a DLT, the cohort was to be expanded to 12 patients and this was to be considered the recommended phase 2 dose (RP2D) if ≤4/12 patients had DLT events. For phase 2 of the CLAG-M/GO trial, a two-stage design was used to evaluate the primary endpoint of 6-month event-free survival (EFS). The null (historical) 6-month EFS rate was assumed to be 68% based on the experience in the phase 1/2 CLAG-M trial patient population [9]; the study was powered to evaluate a 6-month EFS of 81%. The first stage analysis occurred after 30 patients were treated at the RP2D, and trial continuation was allowed if ≥20 patients were alive without event at 6 months after study registration. If 46 or more of the 60 patients treated at the RP2D were alive and without event at 6 months after study registration, the regimen was considered of interest for further investigation. This design had a one-sided type-1 error rate of 13% and a power of 95%. If 45 or fewer patients treated at the RP2D had an EFS event before 6 months after study registration, 1-year EFS would be additionally evaluated, motivated by prior research with GO showing late-emerging benefits of the addition of GO. If 42 or more patients were alive without event at 1 year, the regimen would be considered of interest for further investigation. Wilcoxon rank sum and Fisher’s exact tests were used to evaluate association with quantitative and categorical variables. For CLAG-M/GO and CLAG-M cohorts, unadjusted probabilities of EFS (measured from day 1 of the first treatment cycle; events = failure to achieve CR/CRi, relapse from CR/CRi, and death), relapse-free survival (RFS, measured date of CR/CRi; events = relapse and death), and overall survival (OS, measured from day 1 of the first treatment cycle; event = death) were estimated using the Kaplan-Meier method, and the probability of time to relapse (TTR, measured from the day of CR/CRi achievement) was summarized using cumulative incidence estimates. Multivariate logistic models (for CR), Cox regression models (for EFS and OS), and cause-specific hazard models (time to relapse) were used to compare outcomes between CLAG-M + GO and CLAG-M. The data cut-off date for analysis was 15 January 2022.

## 3. Results

### 3.1. Study Cohort and Treatment

Between August 2018 and October 2020, our trial enrolled 66 patients with a median age of 65 (range: 19–80) years and a median TRM score of 3.5 (range: 0.02–11.8). Baseline patient- and disease-related characteristics are summarized in Table 1.

Fifty-three (80%) patients had AML, nine (14%) had MDS with excess blasts-2 (MDS-EB-2), three (5%) had chronic myelomonocytic leukemia-2, and one (2%) had another high-grade myeloid neoplasm; twenty-seven (41%) had secondary disease. Using the ELN 2017 cytogenetic/molecular risk criteria [14], twenty-two (33%) patients had favorable risk disease, fifteen (23%) had intermediate risk disease, and twenty-nine (44%) had adverse risk disease. All patients received at least one course of therapy: 29 received one, 26 received two, 6 received three, and 5 received four courses of study therapy. Of the 29 patients receiving only one course of therapy, 4 (14%) did not receive the full planned chemotherapy course because of adverse events (hypoxia, hemorrhage, neurotoxicity, and tumor lysis syndrome). Seventeen of the sixty-six (26%) patients were removed from protocol to undergo allogeneic HCT, and twenty-seven out of sixty-six (41%) received alternative forms of post-remission therapy (many of which were given prior to allogeneic HCT) including high-dose cytarabine as single agent, HMAs, or investigational agent(s). As of the data cut-off, 28 of the 66 patients (42%) underwent allografting.

### 3.2. Phase 1

In phase 1, 18 patients were enrolled and subjected to a median of 1.5 (range: 1–4) cycles of study therapy (Table 1). Because only 1 DLT occurred among the six patients treated at dose level 1 (Table 2), twelve subsequent patients were treated with CLAG-M/GO3. Three DLTs occurred at this second dose level for a DLT rate of 25%, less than the protocol specified acceptable DLT rate of 33%. With this, the MTD was not reached and CLAG-M/GO3 declared the RP2D.

Among the 18 patients treated in phase 1, 12 achieved CR and 2 CRi for a CR rate of 67% (95% exact confidence interval: 41–87%) and a CR/CRi rate of 78% (52–94%). At the time of best response, 10 of the 12 CR patients and 12 of the 14 CR/CRi patients were negative for MRD by MFC and cytogenetics, for an MRD^neg^ CR rate of 56% (31–78%) and an MRD^neg^ CR/CRi rate of 67% (41–87%). Of the four patients without CR or CRi after two cycles of therapy, two underwent allogeneic HCT in aplasia or morphologic leukemia-free state (MLFS); only one patient had resistant disease. One AML patient with marrow involvement and lytic bone lesions achieved an MRD^neg^ marrow remission, but bone lesions were not re-assessed at the time of response and was therefore considered unevaluable for response assessment (Table 2). Eight-week mortality among phase 1 patients was 0%.

### 3.3. RP2D Cohort

Sixty patients received CLAG-M/GO3 (12 patients treated in phase 1 and 48 treated in phase 2, respectively). Amongst these were four patients with an antecedent hematologic disorder who received prior therapy with hypomethylating (HMA) agents. Summarized in Table 3 are the best responses achieved after 1–2 cycles of induction chemotherapy for the entire study population and, separately, those treated at the RP2D.

Within the RP2D cohort, 38 patients achieved an MRD^neg^ CR (63% [50–75%]). Seven patients achieved an MRD^pos^ CR, and seven additional patients a CRi (all MRD^neg^) for a CR/CRi rate of 87% (75–94%). Forty-five of the 52 responders (87%) had no evidence of MRD by multiparameter flow cytometry, for an overall MRD^neg^ CR/CRi rate of 75% (62–85%). Two patients obtained an MLFS (both MRD^neg^), three patients were found to have resistant disease, and one patient died from indeterminate cause. In the RP2D cohort, 8-week mortality was 0%.

For the entire RP2D study population, the cumulative incidence of relapse and estimates of EFS and OS are depicted in Figure 1.

With a median follow-up among censored patients of 22.5 months, estimates of 6- and 12-month relapse rates were 18% (12–23%) and 27% (21–33%). Six- and twelve-month estimates were 73% (63–85%) and 58% (47–72%) for EFS and 90% (83–98%) and 76% (66–89%) for OS, respectively (Table 4). Forty-four of the sixty patients [73%] were alive and without event at 6-months (*p* = 0.20 accounting for the two-stage design), and 32 [53%] were alive and without event at 12-months. Thus, the protocol-defined primary endpoint was not reached.

### 3.4. Grade 3–4 Toxicities and Sinusoidal Obstruction Syndrome/Veno-Occlusive Disease (SOS/VOD)

Besides infections and neutropenic fever, hypertension, cardiomyopathy, elevations of alanine and aspartate aminotransferase levels, tumor lysis syndrome, and pulmonary edema were the most common adverse events with CLAG-M/GO (Table 5).

No grade 5 toxicities were encountered during this trial. SOS/VOD is of particular concern for patients treated with GO [20,21], particularly when undergoing HCT. Following the initial cycle of induction therapy, one patient developed severe, self-limited liver toxicity characterized by weight gain and grade 4 elevation of aminotransferases without hyperbilirubinemia that did not meet modified Seattle criteria for SOS/VOD [22]. Of the 28 patients who underwent allogeneic HCT following study therapy, one (4%) was diagnosed with non-fatal post-HCT SOS/VOD not requiring defibrotide therapy.

### 3.5. Duration of Cytopenias

To minimize confounding by residual leukemia, we assessed the duration of cytopenias following CLAG-M/GO in the subset of 45 RP2D patients achieving a morphologic CR after the first course of induction therapy. Among these patients, the median times to neutrophil recovery to 1000/µL and platelet recovery to 100,000/µL were 32 (range: 22–51) days and 31 (range: 21–48) days.

### 3.6. Treatment Outcomes in Various Patient Subgroups

In our phase 1/2 trial of CLAG-M monotherapy, cytogenetic disease risk was strongly associated with response and outcome [9]. This was also the case in this CLAG-M/GO trial, both regarding response rates and estimates of relapse rates and survival (Table 4). In multivariable analyses, intermediate- and adverse-risk groups exhibited decreased odds of a CR compared to the favorable-risk group (odds ratio [OR] = 0.80, *p* = 0.003; and OR = 0.74, *p* < 0.001). Hazard ratios (HRs) for EFS and OS were 2.32 (*p* = 0.004) and 1.96 (*p* = 0.048) for intermediate-risk disease and 3.56 (*p* < 0.001) and 3.49 (*p* < 0.001) for adverse-risk disease, respectively.

In older patients, potential risks of intensive regimens could outweigh potential benefits. We therefore performed subset analyses in which we compared response rates and tolerability of CLAG-M/GO3 between older adults (≥65 years of age; *n* = 32) and younger adults (<65 years; *n* = 28). As one might expect, older patients had higher baseline TRM scores, presented more frequently with secondary disease and/or adverse-risk cytogenetics, and were less likely to receive allogeneic HCT following study therapy (Table 1). Response rates were lower in older patients: 85% vs. 65% (CR), 89% vs. 84% (CR/CRi), and 71% vs. 56% (MRD^neg^ CR; Table 3). Estimates of relapse rates, EFS, OS are summarized in Table 4 and depicted in Figure 1.

### 3.7. Comparison of CLAG-M/GO3 vs. CLAG-M

To put our trial results into context, we retrospectively compared outcomes of the 60 patients treated with CLAG-M/GO3 with 186 medically matched adults with AML or another high-grade myeloid neoplasm given CLAG-M as upfront therapy between December 2014 and January 2021. For medical fitness matching, patients treated with CLAG-M were required to have a TRM score of ≤13.1, serum creatinine ≤2.0 mg/dL, total bilirubin ≤2.5 times the upper limit of normal, and LVEF ≥45% if cardiac function measurements were available (LVEF data were missing in 24 patients). There were no statistically significant differences between the CLAG-M/GO3 and CLAG-M cohorts regarding age, disease type, cytogenetic/molecular risk, proportion of secondary disease (numerically higher in CLAG-M/GO3), and TRM scores (Table 6).

The CR/CRi rate was numerically higher after CLAG-M/GO3 without reaching statistical significance (87% vs. 78%; *p* = 0.19). Among patients in CR after a first cycle of chemotherapy, the median time to neutrophil recovery to 1000/µL and platelet recovery to 100,000/µL was 4–5 days longer with CLAG-M/GO3 (*p* ≤ 0.001). While the 4-week mortality rate was similar (0% vs. 3%; *p* = 0.34), the 8-week mortality rate was lower with CLAG-M/GO3 (0% vs. 7%; *p* = 0.042). Across all patients, estimates for EFS and OS were longer following CLAG-M/GO3 than CLAG-M after adjustment for TRM scores (for EFS: HR = 0.68 [0.45–1.01], *p* = 0.054; for OS: HR = 0.58 [0.37–0.92], *p* = 0.020), whereas cumulative incidences of relapse were similar (HR = 0.93 [0.54–1.59], *p* = 0.78; Figure 2). We then evaluated whether the benefit of CLAG-M/GO3 varied across ELN 2017 cytogenetic/molecular risk groups. In the subset of 73 patients with favorable-risk disease, CLAG-M/GO3 was associated with significantly better EFS (HR = 0.14 [0.03–0.59], *p* = 0.007) and OS (HR = 0.19 [0.04–0.86], *p* = 0.030) and a trend toward lower relapse risks (HR = 0.24 [0.05–1.09], *p* = 0.064) (Figure 3). In contrast, there were no statistically significant differences in relapse rates, EFS, or OS between CLAG-M/GO and CLAG-M among the 41 patients with intermediate-risk disease or the 101 patients with adverse-risk disease, respectively.

## 4. Discussion

The add-on of GO to upfront intensive AML chemotherapy has come of age. An important lesson from the several thousand adults with newly diagnosed AML participating in randomized controlled trials is that GO at doses of 3 mg/m^2^ provides a benefit in this situation for at least a subset of patients [1,2,3,4,5,6,7,8]. However, the safety window is narrow, and GO already in doses of 6 mg/m^2^ is associated with increased toxicity and no benefit when used with intensive chemotherapy [8,23]. Even GO doses of 3 mg/m^2^ can be problematic: as a recent example, in the AMLSG 09–09 trial, a single 3 mg/m^2^ dose of GO added to double-induction chemotherapy with idarubicin, standard-dose cytarabine, etoposide and *all*-trans retinoic acid nearly doubled the rate of deaths during induction compared to the intensive chemotherapy backbone alone (10.3% vs. 5.7%) [6]. Thus, both efficacy and safety need to be carefully assessed before GO is routinely used with a new chemotherapy backbone.

The primary objective of our trial was to test whether GO3, i.e., 3 mg/m^2^ of GO on days 1, 4, and 7, could be safely added to a chemotherapy backbone consisting of CLAG-M with escalated doses of mitoxantrone. While a single dose of GO at 3 mg/m^2^ has been successfully used with fludarabine/high-dose cytarabine/G-CSF/idarubicin (FLAG-Ida) [1], a backbone similar to CLAG-M, data describing whether GO3 could be combined with high-dose cytarabine-containing AML chemotherapy were lacking. Though our trial does not address whether GO3 is superior to GO1 when combined with high-dose cytarabine-containing chemotherapy, our findings presented herein, which use CLAG-M as the high-dose cytarabine-containing chemotherapy backbone, suggest that the combination with GO3 is indeed feasible with manageable toxicities and low early mortality (in our trial: 8-week mortality rate of 0%) in medically fit adults. Approximately 50% of our trial participants were 65 years of age or older, indicating that CLAG-M/GO3 can also be given relatively safely to older, fit adults, with fitness determined by a low TRM score [12]. Of importance, our data indicate a relatively low rate of significant liver toxicity with this combination therapy. One patient had reversible early liver injury not meeting SOS/VOD criteria, and only one patient who underwent allogeneic HCT following study therapy was diagnosed with non-fatal post-HCT SOS/VOD. Considering that half of the patients ≤65 years and one third of patients >65 years underwent allogeneic HCT following CLAG-M/GO3, this therapy appears safe even in patients subsequently receiving allografts.

To better address the important question of whether the addition of GO improves outcomes with CLAG-M chemotherapy, we compared outcomes with CLAG-M/GO3 with outcomes observed in 186 medically matched patients treated with CLAG-M, corresponding to an updated and expanded patient cohort relative to the cohort of patients treated on our previously reported phase 1/2 trial [9]. While limited because of its non-randomized nature, this comparison showed that CLAG-M/GO was associated with achievement of a non-significantly higher CR/CRi rate, lower 8-week (but not 4-week) mortality despite a prolongation of platelet and neutrophil recovery by 4 days (acknowledging the overall rarity of early death events), and better EFS and OS relative to CLAG-M. Considering that previous randomized controlled trials found survival benefit from GO add-on primarily in patients with favorable-risk disease [8], we asked whether effects were similar across different cytogenetic/molecular risk categories. We found that survival improvements with the addition of GO3 to CLAG-M were limited to patients with cytogenetically/molecularly favorable-risk disease. This observation largely explains why, across the entire study cohort, the primary endpoint of the phase 2 portion of the study was not reached. Our finding of significantly improved survival limited to favorable risk patients very closely parallels the experience from five of the randomized trials of GO add-on to intensive chemotherapy. In these trials, GO did not significantly increase the proportion of patients achieving CR or CRi but did improve survival estimates, with the largest benefits in patients with favorable-risk disease [8]. With CLAG-M/GO3, we observed a very low rate of relapse in favorable-risk patients (7% at 6 months, 14% at 12 months), with EFS and OS estimates of 95% and 100% at 1 year.

## 5. Conclusions

Considering CLAG-M/GO3 appears safe in medically fit adults, our findings suggest that this regimen should be considered for adults with AML or other high-grade myeloid neoplasms and favorable disease-risk features.

## Figures and Tables

**Figure 1 cancers-14-02934-f001:**
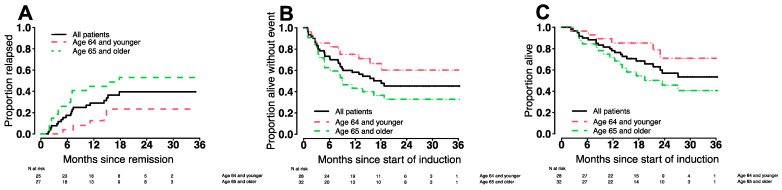
Estimates of (**A**) cumulative incidence of relapse, (**B**) event-free survival, and (**C**) overall survival for the 60 patients who received CLAG-M/GO3 as well as, separately, the 28 patients age <65 years and the 32 patients age ≥65 years in this cohort.

**Figure 2 cancers-14-02934-f002:**
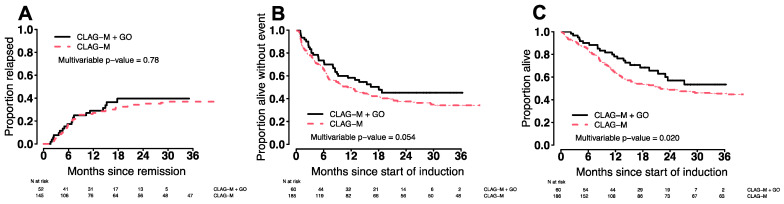
Estimates of (**A**) cumulative incidence of relapse, (**B**) event-free survival, and (**C**) overall survival for the 60 patients who received CLAG-M/GO3 and 186 medically matched patients who received CLAG-M without GO.

**Figure 3 cancers-14-02934-f003:**
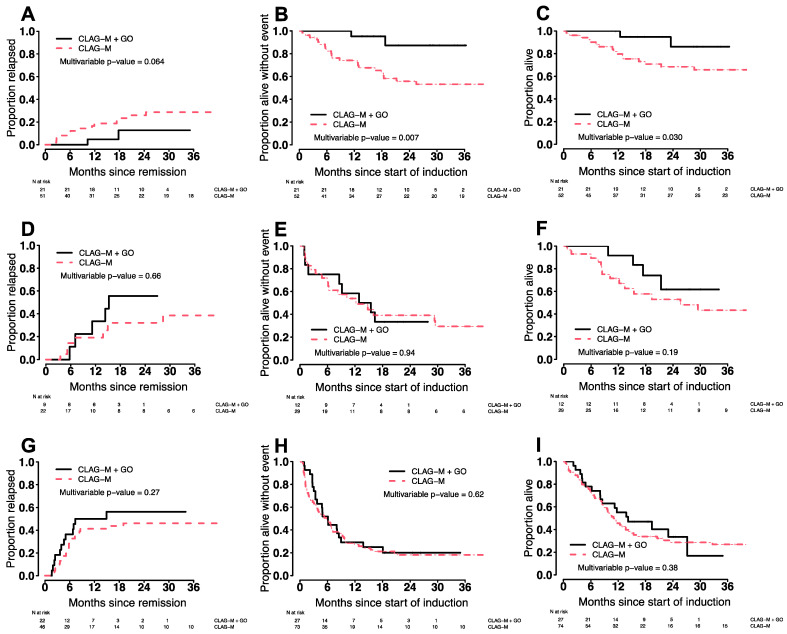
Estimates of (**A**,**D**,**G**) cumulative incidence of relapse, (**B**,**E**,**H**) event-free survival, and (**C**,**F**,**I**) overall survival for the 60 patients who received CLAG-M/GO3 and medically matched patients who received CLAG-M without GO, stratified by favorable (**A**–**C**), intermediate (**D**–**F**), and adverse (**G**–**I**) cytogenetic/molecular disease risk (ELN 2017 classification).

**Table 1 cancers-14-02934-t001:** Characteristics of CLAG-M/GO study cohort.

Parameter	All Patients (*n* = 66)	Phase 1 Cohort (*n* = 18)	RP2D Cohort (*n* = 60)	RP2D, Age < 65 (*n* = 28)	RP2D, Age ≥ 65 (*n* = 32)
Age, median (range), years	65 (19–80)	66 (29–78)	65 (19–80)	51 (19–64)	71 (65–80)
Male gender, *n* (%)	34 (52%)	6 (33%)	32 (53%)	17 (61%)	15 (47%)
Disease					
AML	53 (80%)	14 (78%)	48 (80%)	22 (85%)	26 (81%)
With recurrent genetic abnormalities	25 (38%)	5 (28%)	23 (38%)	11 (39%)	12 (38%)
With myelodysplasia-related changes	14 (21%) ^‡^	4 (22%) ^‡^	11 (18%)	3 (11%)	8 (25%)
Treatment-related myeloid neoplasm	5 (8%)	3 (17%)	5 (8%)	2 (7%)	3 (9%)
AML, not otherwise specified	9 (14%)	2 (11%)	9 (15%)	6 (21%)	3 (9%)
MDS-EB-2	9 (14%)	4 (22%)	8 (13%)	4 (14%)	4 (13%)
CMML-2	3 (5%)	0 (0%)	3 (5%)	2 (7%)	1 (3%)
Other high-grade myeloid neoplasm	1 (2%)	0 (0%)	1 (2%)	0 (0%)	1 (3%)
Secondary disease *	27 (41%)	8 (44%)	24 (40%)	6 (21%)	18 (56%)
Median TRM score (range)	3.5 (0.02–11.8)	3.9 (0.14–10.4)	3.5 (0.02–11.8)	1.8 (0.14–9)	4.4 (0.02–11.8)
Performance status, *n* (%)					
0	14 (21%)	3 (17%)	12 (20%)	9 (32%)	3 (9%)
1	45 (68%)	14 (78%)	42 (70%)	17 (61%)	25 (78%)
2	7 (11%)	1 (6%)	6 (10%)	2 (7%)	4 (13%)
Cytogenetic/molecular risk, *n* (%) **					
Favorable	22 (33%)	7 (39%)	21 (35%)	12 (43%)	9 (28%)
Intermediate	15 (23%)	5 (28%)	12 (20%)	7 (25%)	5 (16%)
Adverse	29 (44%)	6 (33%)	27 (45%)	9 (32%)	18 (56%)
Mutational status, *n* (%)					
*FLT3*-ITD					
Wild type	54 (82%)	15 (83%)	49 (82%)	24 (86%)	25 (78%)
Mutated	7 (11%)	1 (6%)	7 (12%)	2 (7%)	5 (16%)
Unknown	5 (8%)	2 (11%)	4 (7%)	2 (7%)	2 (6%)
*NPM1*					
Wild type	44 (67%)	12 (67%)	39 (65%)	19 (68%)	20 (63%)
Mutated	17 (26%)	4 (22%)	17 (28%)	7 (25%)	10 (31%)
Unknown	5 (8%)	2 (11%)	4 (7%)	2 (7%)	2 (6%)
*TP53*					
Wild type	46 (70%)	13 (72%)	42 (70%)	21 (75%)	21 (66%)
Mutated	8 (12%)	2 (11%)	7 (12%)	4 (14%)	3 (9%)
Unknown	12 (18%)	3 (17%)	11 (18%)	3 (11%)	8 (25%)
*RUNX1*					
Wild type	46 (70%)	13 (72%)	42 (70%)	23 (82%)	19 (59%)
Mutated	8 (12%)	2 (11%)	7 (12%)	2 (7%)	5 (16%)
Unknown	12 (18%)	3 (17%)	11 (18%)	3 (11%)	8 (25%)
Laboratory findings at screening, median (range)					
Total WBC count (×10^9^/L)	6.6 (0.8–157)	6.3 (0.8–157)	6.9 (0.8–157)	7.2 (0.8–157)	6.1 (1.3–64)
Absolute neutrophil count (×10^9^/L)	1.2 (0–23)	1.1 (0.03–6.9)	1.2 (0–23)	1.2 (0–20)	1.2 (0.1–23)
Peripheral blood blasts (%)	21 (0–94)	27 (0–87)	23 (0–94)	28 (0–90)	20 (0–94)
Hemoglobin (g/dL)	8.9 (6.5–13.3)	9.2 (6.8–13.2)	8.9 (6.5–13.3)	8.8 (6.8–11.9)	8.9 (6.5–13.3)
Platelets (×10^9^/L)	59 (6–820)	50 (12–413)	62 (6–820)	50 (8–413)	77 (6–820)
Serum creatinine (mg/dL)	0.8 (0.5–1.8)	0.8 (0.5–1.7)	0.8 (0.5–1.8)	0.7 (0.5–1.2)	0.9 (0.5–1.8)
Total bilirubin (mg/dL)	0.5 (0.3–1.7)	0.5 (0.3–1.7)	0.5 (0.3–1.7)	0.4 (0.3–1.7)	0.6 (0.3–1.2)
AST (U/L)	18 (9–72)	17 (11–72)	18 (9–72)	17 (9–72)	20 (10–39)
ALT (U/L)	15 (5–138)	13 (5–120)	16 (5–138)	16 (8–138)	16 (5–42)
Subsequent allogeneic HCT	28 (42%)	8 (44%)	25 (42%)	14 (50%)	11 (34%)

* Defined either as AML transformed from antecedent hematologic disorder, high-grade myeloid neoplasm other than AML if there was a history of lower-grade myeloid neoplasm (i.e., <10% blasts in blood and marrow), or AML/MDS in a patient who had previously received cytotoxic therapy or radiation. ** ELN 2017 criteria. ^‡^ One patient with both myelodysplasia-related changes and myeloid sarcoma. Abbreviations: RP2D, recommended phase 2 dose; TRM, treatment-related mortality; WBC, white blood cell.

**Table 2 cancers-14-02934-t002:** Dose escalation scheme, best responses, and dose-limiting toxicities during phase 1, *n* = 18.

Dose Level	GO (D0 to D5)	Patients (*n*)	Best Response	Dose-Limiting Toxicities
1 (GO1)	3 mg/m^2^ on day 1	6	1 unevaluable * 1 CRi MRD^neg^ 4 CR MRD^neg^	1 grade 3 left ventricular systolic dysfunction
2 (GO3)	3 mg/m^2^ on days 1, 4, and 7	12	6 CR MRD^neg^ 2 CR MRD^pos^ 1 CRi MRD^neg^ 1 MLFS MRD^neg^ 1 Aplasia ** MRD^neg^ 1 RD	1 grade 4 aminotransferase level increase 1 grade 3 posterior reversible encephalopathy syndrome 1 grade 3 intracranial hemorrhage

Abbreviations: CR, complete remission; GO, gemtuzumab ozogamicin; MRD, measurable residual disease; CRi, complete remission with incomplete hematologic recovery; MLFS, morphologic leukemia-free state; RD, resistant disease. * One patient with MRD^neg^ marrow assessment but no follow-up imaging assessment for extramedullary disease (lytic bone lesions); ** patient with resistant disease after Cycle 1 who did not have a bone marrow assessment prior to death after Cycle 2 reinduction.

**Table 3 cancers-14-02934-t003:** Best response after 1–2 cycles of study therapy.

Response, *n* (%)	All Patients (*n* = 66)	RP2D Cohort (*n* = 60)	RP2D, Age < 65 (*n* = 28)	RP2D, Age ≥ 65 (*n* = 32)
CR				
MRD^neg^	42 (64%)	38 (63%)	20 (71%)	18 (56%)
MRD^pos^	7 (11%)	7 (12%)	4 (14%)	3 (9%)
CRi				
MRD^neg^	8 (12%)	7 (12%)	1 (4%)	6 (19%)
CR/CRi	57 (86%)	52 (87%)	25 (89%)	27 (84%)
MLFS				
MRD^neg^	2 (3%)	2 (3%)	1 (4%)	1 (3%)
Aplasia				
MRD^neg^	2 (3%)	2 (3%)	1 (4%)	1 (3%)
Resistant disease	3 (5%)	3 (5%)	1 (4%)	2 (6%)
Death from indeterminate cause	1 ^‡^ (2%)	1 ^‡^ (2%)	0	1 (3%)
Unevaluable for response	1 ^§^ (2%)	0 (0%)	0 (0%)	0 (0%)
8-week mortality	0 (0%)	0 (0%)	0 (0%)	0 (0%)

^‡^ Patient with resistant disease after cycle 1 who did not have a bone marrow assessment prior to death after cycle 2 reinduction. ^§^ Patient with myeloid sarcoma who obtained an MRD^neg^ marrow, but extramedullary disease (lytic bone lesions) was not re-assessed at the time of response. Abbreviations: RP2D, recommended phase 2 dose; HMA, “hypomethylating” agents (i.e., azanucleosides); CR, complete remission; MRD, measurable residual disease; CRi, complete remission with incomplete hematologic recovery; MLFS, morphologic leukemia free state.

**Table 4 cancers-14-02934-t004:** Outcome probabilities (with 95% confidence interval) in entire RP2D cohort and stratified by age or cytogenetic/molecular disease risk.

	Overall Survival		Event-Free Survival		Cumulative Incidence of Relapse	
	6 Months	12 Months	6 Months	12 Months	6 Months	12 Months
All patients	90% (83–98%)	76% (66–89%)	73% (63–85%)	58% (47–72%)	18% (12–23%)	27% (21–33%)
By age						
<65 years	96% (84–100%)	85% (73–100%)	86% (74–100%)	75% (61–93%)	16% (10–23%)	21% (14–29%)
≥65 years	84% (73–98%)	72% (58–89%)	63% (48–82%)	43% (29–65%)	20% (12–28%)	34% (24–44%)
By disease risk *						
Favorable	100%	100%	100%	95% (87–100%)	7% (3–15%)	14% (7–24%)
Intermediate	100%	92% (78–100%)	75% (54–100%)	58% (36–94%)	12% (4–26%)	22% (9–38%)
Adverse	78% (64–95%)	55% (39–78%)	52% (36–75%)	29% (16–53%)	32% (21–43%)	44% (32–55%)

* ELN 2017 criteria.

**Table 5 cancers-14-02934-t005:** Grade 3–4 treatment emergent adverse events (TEAEs) occurring with CLAG-M/GO during the first treatment cycle in patients treated in phase 1 and at the RP2D.

Adverse Events by System Organ Class	Phase 1 Cohort (*n* = 18)	RP2D Cohort (*n* = 60)
Blood and lymphatic system disorders *		
Disseminated intravascular coagulation	−	4 (7%)
Febrile neutropenia	15 (83%)	50 (83%)
Infections and infestations		
Catheter-related infection	−	3 (5%)
Encephalitis	−	1 (2%)
Enterocolitis	2 (11%)	1 (2%)
Lung infection	1 (6%)	7 (12%)
Sepsis	6 (33%)	23 (38%)
Skin infection	−	1 (2%)
Urinary tract infection	1 (6%)	2 (3%)
Cardiac disorders		
Atrial fibrillation	−	1 (2%)
Cardiac arrest	−	1 (2%)
Cardiomyopathy	3 (17%)	5 (8%)
Edema	−	1 (2%)
Hypertension	2 (11%)	4 (7%)
Hypotension	1 (6%)	−
Pericardial effusion	−	1 (2%)
Myocardial infarction	−	1 (2%)
Gastrointestinal disorders		
Colonic perforation	−	1 (2%)
Dysphagia	−	2 (3%)
Esophagitis	−	1 (2%)
Gastrointestinal hemorrhage	1 (6%)	5 (8%)
Mucositis	−	3 (5%)
Vomiting	1 (6%)	1 (2%)
Investigations		
Alanine aminotransferase increased	2 (11%)	3 (5%)
Aspartate aminotransferase increased	2 (11%)	5 (8%)
Blood bilirubin increased	−	1 (2%)
Metabolism and nutrition disorders		
Acidosis	−	1 (2%)
Anorexia	−	1 (2%)
Hyperglycemia	−	4 (7%)
Hyperphosphatemia	−	1 (2%)
Hypocalcemia	−	2 (3%)
Hypokalemia	1 (6%)	1 (2%)
Hypophosphatemia	−	1 (2%)
Tumor lysis	2 (11%)	6 (10%)
Nervous system disorders		
Ataxia	−	1 (2%)
Dysarthria	−	1 (2%)
Intracranial hemorrhage	1 (6%)	2 (3%)
Reversible posterior encephalopathy syndrome	1 (6%)	1 (2%)
Stroke	−	1 (2%)
Syncope	−	1 (2%)
Renal and urinary disorders		
Acute kidney injury	−	1 (2%)
Respiratory, thoracic, and mediastinal disorders		
Hypoxia	−	4 (7%)
Pulmonary edema/effusion	1 (6%)	5 (8%)
Pulmonary hemorrhage/hemoptysis	−	2 (3%)
Respiratory failure	−	2 (3%)
Skin and subcutaneous tissue disorders		
Maculo-papular rash	−	1 (2%)

Table describing number of individual patients (% of cohort) affected by Grade 3–4 (CTCAE 5.0) treatment-emergent adverse events considered definitively, probably, or possibly related to study treatment by the investigator that were experienced in the first treatment cycle by the phase 1 cohort and the 60 patients treated at the RP2D. There were no Grade 5 AEs recorded during the study following the first treatment cycle. * Per protocol, hematologic TEAEs were not collected, except for febrile neutropenia and disseminated intravascular coagulation.

**Table 6 cancers-14-02934-t006:** Comparison of CLAG-M/GO3 and CLAG-M cohorts.

Regimen	CLAG-M/GO3 (*n* = 60)	CLAG-M (*n* = 186)	*p*-Value
Median age, years (range)	65 (19–80)	64 (19–84)	0.18
Disease, *n* (%)			0.49
AML	48 (80%)	158 (85%)	
MDS-EB-2	8 (13%)	21 (11%)	
Other	4 (7%)	7 (4%)	
Secondary disease *, *n* (%)	24 (40%)	54 (29%)	0.15
Disease risk (ELN2017), *n* (%)			0.94
Favorable	21 (35%)	52 (34%)	
Intermediate	12 (20%)	29 (19%)	
Adverse	27 (45%)	74 (48%)	
Insufficient data	0	31	
TRM score (range)	3.5 (0.02–11.8)	3.0 (0.01–12.3)	0.54
Performance status, *n* (%)			0.51
0	12 (20%)	45 (24%)	
1	42 (70%)	129 (69%)	
2	6 (10%)	10 (5%)	
3	0 (0%)	2 (1%)	
CR, *n* (%)			1
MRD^neg^	38 (63%)	116 (62%)	
MRD^pos^	7 (12%)	20 (11%)	
CRi, *n* (%)			0.21
MRD^neg^	7 (12)	6 (3%)	
MRD^pos^	0 (0%)	3 (2%)	
CR/CRi, *n* (%)	52 (87%)	145 (78%)	0.19
MLFS or marrow aplasia, *n* (%)	4 (7%)	15 (8%)	1
Resistant disease, *n* (%)	3 (5%)	17 (9%)	0.42
Death from indeterminate cause, *n* (%)	1 (2%)	9 (5%)	0.46
Early mortality, *n* (%)			
4-week	0 (0%)	6 (3%)	0.34
8-week	0 (%)	13 (7%)	0.042
Days to ANC ≥1000/µL, median (range) **	32 (22–51)	28 (18–60)	0.001
Days to platelets ≥100,000/µL, median (range) **	31 (21–48)	26 (18–54)	<0.001
Subsequent allogeneic HCT within 2 years, *n* (%)	25 (42%)	94 (51%)	0.24

* Defined either as AML transformed from antecedent hematologic disorder, high-grade myeloid neoplasm other than AML if there was a history of lower-grade myeloid neoplasm (i.e., <10% blasts in blood and marrow), or AML/MDS in a patient who had previously received cytotoxic therapy or radiation. ** Induction 1; patients achieving CR only. Abbreviations: AML, acute myeloid leukemia; ANC, absolute neutrophil count; CR, complete remission; CRi, complete remission with incomplete hematopoietic recovery; HCT, hematopoietic cell transplantation; MDS-EB-2, myelodysplastic syndrome with excess blasts 2; MLFS, morphologic leukemia-free state; MRD, measurable residual disease; RP2D, recommended phase 2 dose; TRM, treatment-related mortality.

## Data Availability

For original, de-identified data, please contact the corresponding author.

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
