# Peer review of "Phase 1/2 Trial of CLAG-M with Dose-Escalated Mitoxantrone in Combination with Fractionated-Dose Gemtuzumab Ozogamicin for Newly Diagnosed Acute Myeloid Leukemia and Other High-Grade Myeloid Neoplasms"

_cancers, 2022, doi:10.3390/cancers14122934_

Round 1

Reviewer 1 Report

This is a well written manuscript on a non-randomized, historically controlled, clinical trial. There are more published clinical trials on the addition of GO to a high dose chemo, some of which are randomized. The combination of CLAG-M with GO was not studied before.

The study shows that fractionated GO 3 mg/m2 x 3 can safely be given to adult and even elderly (fit) AML patients. Only favorable risk AML patients seem to clinically benefit, as was previously shown in other clinical trials with a different chemotherapy backbone.

This therefore adds some new information to what is already known.

I do have some questions:

What proportion of adult AML pts are excluded by only including 'fit' participants (TRM score of ≤13.1)?

You describe that the historical control cohort was 'medically matched'. Please describe how this matching was performed (based on which characteristics)

The neutropenic and neutropenic period was longer in GO treated pts and the proportion of patients with a CRi compared to CR was higher. Was there a difference in a relapse risk between patients achieving a MRD negative CR vs a MRD negative CRi?

You describe that the 12 months relapse risk was 27%, while the 12 m EFS was 48%. Therefore 52% of patients had an event, while only 27% relapsed. What were the other 15% events?

Reviewer 2 Report

The authors compőare the CLAG-M protocol with the GO added CLAG-M. This is a very important trial, as this patient population is having a suboptimal response to first line therapy. The results are well documented, very important. I recommend this paper to be published without modifications.
